# A Collaborative Framework for Hydropower Development and Sustainable Livelihood of Farmers in the Lancang-Mekong River Basin: A Review with the Perspective of Energy-Water-Food Nexus

**Shuai Zhong** [1,2,3,4,*], **Yidong Zhu** [1,2], **Jianan Zhao** [1,2] and **Lei Shen** [1,2,4,*]

1    Institute of Geographical Sciences and Natural Resources Research, Chinese Academy of Sciences, Beijing 100101, China; zhuyidong21@mails.ucas.ac.cn (Y.Z.); zhaoja@igsnrr.ac.cn (J.Z.)
2    School of Resources and Environment, University of Chinese Academy of Sciences, Beijing 100049, China
3    Collaborative Innovation Center for Geopolitical Setting of Southwest China and Borderland Development, Kunming 650500, China
4    Laos-China Joint Research Center for Resources and Environment, Vientiane 7864, Laos
*    Correspondence: zhongshuai@igsnrr.ac.cn (S.Z.); shenl@igsnrr.ac.cn (L.S.)

**Abstract:** With the process of poverty eradication and economic growth, hydropower development becomes increasingly important because of its huge potential advantages in the Lancang-Mekong River Basin. However, the complex topography and rich resource endowments in the Lancang-Mekong River Basin bring a variety of potential risks and uncertainties in hydropower development, which has an important impact on the sustainable livelihood of farmers. There is an urgent need for countries in the Lancang-Mekong River Basin to systematically assess hydropower projects, especially their impact on the sustainable livelihoods of farmers. Based on the systematic analysis of relevant literature, this study established a collaborative framework of hydropower development and farmers' sustainable livelihood, including theoretical framework, indicator system and model structure. The purpose is to explore the interaction mechanism of energy and water resources utilization, food security and sustainable livelihood of farmers in hydropower development. The findings can provide scientific and technological support for the Belt and Road Initiative, poverty reduction and sustainable development in the river basin.

**Keywords:** Lancang-Mekong River Basin; dilemma between hydropower and livelihood; energy-water-food nexus; systematic framework

## 1. Introduction

With the process of poverty eradication and economic growth, the countries in the Lancang-Mekong River Basin are committed to finding more energy sources for development. The local natural resource advantage has prompted the development of hydropower. However, the development of a hydropower project inevitably has an impact on ecology and agricultural production in the Lancang-Mekong River Basin, and thus measuring the impact of hydropower development on the sustainable livelihoods of farmers is rapidly becoming a hot focus. The Lancang-Mekong River Basin (LMRB), with an $8.11 \times 10^5$ km$^2$ catchment area, originates in the Tibetan Plateau (~5000 m above the sea level) and flows through China, Myanmar, Laos, Thailand, Cambodia, and Vietnam. It has a total length of 4350 km, providing a livelihood for more than 72 million people. It is one of the richest fish farms in the world, with more than 1300 species, an important "lifeline" for countries in the LMRB, and the most potential area for hydropower in the world [1–3].

The LMRB is one of the typical poor regions in the world; for example, 21.2% of the population in Laos is vulnerable to multidimensional poverty and 10.0% of the population is living below PPP 1.90 per day in 2017. The contribution of deprivation in the standard

of living to overall multidimensional poverty is 38.8%. While Laos is still in poverty, the proportion of multidimensionally poor people have dropped by nearly half in the past few years from 40.24 in 2012, and the poverty dimension indicator for electricity has dropped from 21.8 to 6.1, indicating that access to electricity is of great significance in lifting Laos out of poverty [4]. At the same time, price fluctuations in the international energy market and the increasing energy demand for the development of the Mekong region have generated strong interest in local hydropower developments [5]. In addition, there is strong interest in the Mekong countries to increase income and solve the problem of energy shortage. Hydropower development is an integral part of Laos and its poverty eradication strategy. The development of hydropower projects not only provides security of local energy supply, but also enables the sale of electricity to neighboring countries [6,7]. According to the development plan of the Royal Government of Cambodia, increasing energy supply is a priority due to the high domestic energy costs and low energy access, and hydropower is considered the main domestic renewable energy option to improve energy security [8]. In addition, Mekong River Basin vigorously developed agriculture and trade in response to food security. Agricultural product trade is an important source of income for the Mekong River Basin. According to 2016 statistics, China has become an indispensable and important partner in the agricultural trade of the Mekong Basin countries [9]. China's agricultural products trade with Mekong River Basin countries amounted to USD 15.84 billion (16.33% of the total agricultural product trade of Mekong River Basin countries and 8.22% of the total agricultural trade of China).

In recent years, the analysis of hydropower development in the LMRB and its impact on resources and the environment, especially on water resources, food security and farmers' livelihood, has become a significant topic. Lancang-Mekong River is the tenth largest river in the world, with an annual runoff of 14,500 $m^3$/s. The population in the LMRB is growing rapidly and is estimated to reach 145 million by 2050, compared to 2015 [10]. Population growth and rapid urbanization have led to a large number of hydropower and reservoir construction projects. By the end of 2016, the hydropower potential in the Mekong Basin was 235,000 GWh per year, and 82 dams were built with a storage capacity of 82 $km^3$ (Figure 1). The share of storage in the basin countries is 51% in China, 33% in Laos, 12% in Thailand, 0.6% in Cambodia and 3% in Vietnam [11]. The number of dams is expected to reach 138 by 2025, and the storage volume reached 21% of the total downstream [12]. At the same time, because of the southwest monsoon, the LMRB has a unique dry season (from November to April) and rainy season (from June to October), with basically no rain in the dry season and up to 80–90% of the annual rainfall in the rainy season, making the water resources in the LMRB highly variable seasonally, with more than 75% recharge in the rainy season, which is highly susceptible to serious flood and drought disasters. Although the construction of dams and reservoirs in the LMRB may mitigate the risk of natural disasters and improve the natural conditions for agricultural and fishery development, it also has a series of impacts on the ecological environment and rural production and life downstream and is highly controversial among all parties [11,13].

Hydropower development in the LMRB may also affect the livelihoods of farmers in the basin due to disturbances of fisheries and land use. With a poverty rate of 19% in the lower Mekong River Basin and 21% of the population without access to clean water, countries have made improving the livelihoods of farmers a development priority, including poverty reduction, development of modern agriculture and disaster prevention and mitigation. The Mekong River Commission's first assessment report on the environmental and social impacts of dams in the LMRB shows that dam construction has high environmental costs resulting in fish reduction, riverbank inundation, soil nutrient loss, direct losses of USD 500 million per year, direct or indirect loss of livelihoods for about 2.1 million people and an average annual loss of about 30% of protein intake in countries that rely on fish as a major food source, such as Laos and Cambodia [14]. At the same time, the increasingly severe climate change situation makes the LMRB face more serious flood and drought risks [15,16]. Expanding the scale of hydropower development must take into account

climate change factors and promote hydropower development and solar, wind, biomass and other renewable energy developments in coordination, can effectively improve the livelihood of the surrounding communities and ecological environment issues [17]. Scientific dam construction and management can ensure and promote the seasonal regulation of water resources supply, agricultural production and fishery production. According to relevant simulation analysis, optimal management of water flow through dams can significantly increase the efficiency of fisheries output by up to 3.7 times [18].

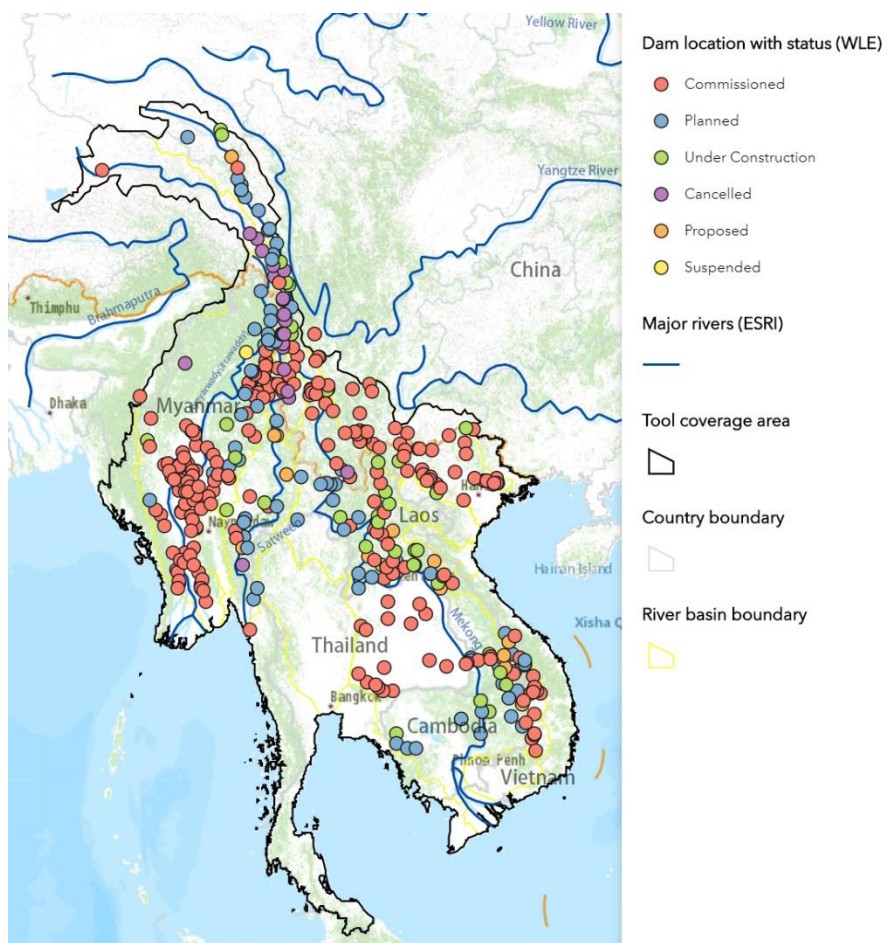

**Figure 1.** Different types of hydropower projects and their distribution in the Lancang-Mekong River Basin. NOTE: the original sources refer to http://damtool-servir.adpc.net/ (assessed on 31 December 2021).

Therefore, the multi-win goal of realizing the benefits of hydropower development in the LMRB and reconciling it with the improvement of farmers' livelihoods is increasingly becoming a focus of the basin countries to advance their sustainable development goals. The key scientific issue is to reveal the mechanism of interaction between hydropower development and sustainable livelihoods of farmers, and the core technical challenge is to identify the energy-water-food (EWF) nexus. The EWF linked process of hydropower development has a complex interaction with the sustainable livelihoods of farm households. For example, hydroelectric power production requires sufficient and stable water potential energy, which requires full consideration of topography, hydrology, droughts, floods, temporal and spatial variations of monsoon precipitation [12]. Dam construction also provides irrigation water for food production and maintains the health of the watershed ecosystem, which inevitably has significant impacts on surrounding land use, aquaculture and farming, power availability and the livelihoods of farmers. Dam construction also provides irrigation water for food production and maintains the ecosystem health of LMRB,

which inevitably has major impacts on surrounding land use, aquaculture and farming, power availability and the livelihoods of farmers.

This study addresses the synergistic benefits of hydropower development and sustainable livelihoods of farm households in the LMRB. It systematically compares relevant studies on hydropower development, EWF nexus analysis and sustainable livelihoods of farm households in the LMRB and proposes a collaborative analysis framework of hydropower development and sustainable livelihoods of farm households in the LMRB based on the EWF nexus. The study intends to address the following three questions:

(1) What are the advantages, characteristics or problems of hydropower development in the LMRB based on the EWF nexus perspective; that is, why to apply the EWF nexus analysis method.

(2) How to achieve synergy between hydropower development and sustainable livelihoods of farmers in the LMRB, what are the interacting indicator relationships and how do the linked processes of energy use, water use and food production function.

(3) What problems will be solved, what advantages will it have and how can it be expanded by applying the collaborative analysis framework of hydropower development and farmers' sustainable livelihood in the LMRB.

## 2. Hydropower Development in the LMRB Basin Amidst Controversy

Although the LMRB has great potential for hydraulic resource development, the degree of development is not high compared to other large rivers in the world. Most of the dams currently built or under construction are concentrated in the tributaries of the basin or account for only a small portion of the annual flow of the river [14,19], and future expansion of development may face more intense controversy. In this section, we explore the negative and positive impacts of hydropower development in the LMRB.

### 2.1. Negative Impacts of Hydropower Development in the LMRB

Studies opposing the expansion of hydropower development have concluded that many dam projects currently being built or planned will have a major impact on the ecological environment and production and living conditions. It has been suggested that dam construction projects on the mainstem of the Lancang River in China will alter water levels and hydrological conditions in the downstream Mekong River, reduce sediment transport, block fish migration [20] and exacerbate the risk of phosphorus and nitrogen pollution of reservoir sediments. In addition, heavy metal elements such as Cd, Mn, Pb and Cu have high concentration levels in pre-dam areas, and the construction of terraced reservoirs that slow down flow velocities and have significant cumulative effects on conductivity and turbidity [21–24] will generate a certain degree of ecological risk to aquatic species, especially indigenous fish of the Lancang River [25]. For example, the operation of the Lancang River step dams has resulted in a significant increase in the density and biomass of macro-benthic communities in the stillwater areas of the Manwan reservoir [23], which can also cause siltation and impede sediment transport to the Mekong Delta [26,27], resulting in tens of millions of people whose livelihoods will be affected [20,28]. Dam construction projects in the LMRB will also exacerbate altered water flow conditions, reduce fisheries output and have serious impacts on local ecosystems and aquatic biodiversity. An assessment project on tributaries of the Mekong River has indicated that the four planned dams (Lower Se San 2, Se Kong 3d, Se Kong 3u, Se Kong 4) would result in the decline of downstream fish biomass by 9.3%, 2.3%, 0.9% and 0.75%, respectively [29]. A water footprint tracking of hydropower plants in the LMRB has found that 38 hydropower plants in the LMRB evaporated 2.861 billion $m^3$ of water per year, of which 2.007 billion $m^3$ (70.1%) were evaporated from 28 hydropower plants outside of China [30]. This has raised doubts about whether future hydropower development can follow the concept of "green hydropower" [30], and hydropower development enterprises and other interests should increase the intensity of ecological compensation to ensure that the net income of farmers is not reduced [31]. Moreover, as the amount of water and sediment in the river decreases,

it leads to sedimentation of the riverbed downstream to accelerate the seawater intrusion, which is not conducive to local farmers' farming [12]. Moreover, the degraded arable land area in the lower Mekong reached 65,352 km$^2$ (36% of the total degraded area) in 2015 due to climate, population and land cover type, and the water quality changes in the Mekong River will exacerbate this degradation rate [32].

*2.2. The Positive Role of Hydropower Development in the LMRB*

Studies supporting the expansion of hydropower development have concluded that the combined scheduling of reservoirs across the LMRB has a significant flood control effect on all major sections of the Mekong River, reducing the 200-year flood to 20–50 years when reservoirs are scheduled in full compliance with flood control objectives [33]. A study on the transboundary ecological effects of Lancang River hydropower development on downstream river sediment processes and their heavy metal distribution found that there is an insufficient scientific basis to attribute the main driver of sediment changes in the Mekong River to the construction of the Lancang River mainstem power station and that the hydropower reservoir area has a significant interception effect on upstream and downstream heavy metal pollutants [34]. For the vegetation in the LMRB, hydropower plants have overall strong temporal stability and good utilization of vegetation cover during and after construction, but there are individual hydropower plant areas with reduced vegetation cover [35,36]. In the LMRB, in view of the characteristics of the Mekong River in the dry and rainy seasons, the dam construction can adjust the water supply level through seasonal water management [37], such as releasing water in the dry season to ensure irrigation and shipping [38,39] and storing water in the rainy season to support flood-fighting and disaster relief [18]. A study based on global irrigation data showed that although the general experience worldwide is that dam construction can promote irrigated agriculture, irrigation technology in the LMRB is significantly underdeveloped, rice production is mostly rain-fed, supplemental irrigation beyond rainfall is mainly used for a few fruits, vegetables and cash crops in the Mekong Delta region of Laos, Thailand, Cambodia, and Vietnam and most of the existing irrigation methods use traditional weir structures and mechanical irrigation [40]. In addition, laying out some combination of solar, wind and storage technologies around dams in the LMRB is safer and cheaper than building dams alone [41]. In terms of impact on the surrounding agricultural environment, although hydropower development will lead to a reduction in nutrients such as phosphorus and nitrogen from upstream, affecting water quality, the growing population in the Mekong region has led to an expanding demand for energy in the region, which, according to relevant studies, will require large-scale investment in power generation infrastructure estimated at about USD 191 billion to 217 billion between 2017 and 2050 [42]. The region's current heavy reliance on fossil energy consumption is likely to lead to the construction of coal-fired power plants in the absence of alternative energy sources, which could lead to emissions of sulfur oxides, nitrogen oxides and particulate matter in the region, as well as significant consumption of local water resources, which could be mitigated by the use of hydropower development.

## 3. Sustainable Livelihood Issues for Farmers in the LMRB and the Impact of Hydropower Development

It is foreseeable that the controversy over the expansion of hydropower development in the LMRB will continue. As the Lancang-Mekong River is an international river spanning many countries, it is difficult for the Mekong countries to reach a broad consensus from their different interests, but controversy does not mean no development, and the Mekong countries still support hydropower development projects that have been scientifically shown to be reasonable. Most of the existing studies focus on parts of areas of the basin, such as the mainstream of the Lancang River or tributaries of the Mekong River, and the impact assessment of hydropower development focuses on fishery output, biodiversity and ecological environment changes, but lacks systematic studies on the impact on farmers'

livelihoods and their coping behaviors, which can hardly provide a sufficient basis for integrated resource management and sustainable development of the basin.

### 3.1. Characteristics and Progress of Farmers' Sustainable Development in the LMRB

Global poverty reduction and climate change response activities are the key contents of current international sustainable development research. The sustainable livelihood of farmers in the LMRB has also attracted extensive attention from the international community. Agricultural development conditions in the LMRB, especially in the Mekong region, are generally backward, with a lack of basic electrification equipment, inadequate transportation and power grid infrastructure, and very vulnerable livelihood conditions for farmers, who have low basic incomes and serious poverty problems, but also lack the ability to earn a living and the environment and channels to obtain diversified incomes.

The main idea of improving the sustainable livelihoods of farm households is to focus not only on increasing the income level of farm households, but also on improving their ability to obtain more income. They need to improve their ability to earn a living, develop diversified income channels and mitigate and improve the vulnerability of farm households to sudden risks in their productive lives. Early livelihood studies focused on poverty reduction and related improvements in the productive life environment in developing countries. As international efforts to reduce poverty continue to expand, especially as global environmental change and the impact of human activities intensify, the concept of "sustainable livelihoods" has emerged, introducing the idea of "resilience to climate change and environmental stresses and shocks", i.e., the ability to maintain or deteriorate livelihood activities of the population in the face of regional environmental sustainability problems, even resulting in resource degradation, desertification, deforestation soil erosion, lowering of the water table and salinization [43]; but also the potential to benefit from adaptation activities, such as reforestation and increased levels of agrobiodiversity [44]. The idea of "sustainable livelihoods" was first introduced by the World Commission on Environment and Development in 1987 when discussing resource ownership, basic needs and rural livelihood security [45]. In 1992, the United Nations Conference on Environment and Development adopted the idea of sustainable livelihoods as a link between socio-economic and environmental issues [46]. In 1995, the World Summit for Social Development (WSSD) in Copenhagen highlighted the significance of sustainable livelihoods for poverty reduction and sustainable development. Sustainable livelihoods research continues to expand, showing multi-objective, multi-level and multidisciplinary intersections [47]. For example, increasing agrobiodiversity will increase the resilience of agricultural production and guarantee better nutrition and health and contribute to the sustainability of farmers' livelihoods [48].

### 3.2. Impact of Hydropower Development in the LMRB on Farmers' Livelihoods

Sustainable livelihood issues for farm households in the LMRB focus on poverty reduction and agricultural and fisheries development, especially on vulnerability and resilience to climate change and related natural disasters such as droughts and floods. A study of 56 counties (districts and cities) in the LMRB found that suitability of human settlements showed a deteriorating trend in the time series and decreasing from south to north in the spatial scale, and further improvement of infrastructure construction became the key to improving the living environment [49]. For farming households in north-central Vietnam, natural disasters such as floods will trap poor households in a cycle of vulnerability, and poor households may have difficulty finding adequate off-farm sources of income during disasters due to insufficient skills [50]. Farmers' livelihood sensitivity is also higher in Vietnam than in Cambodia, Laos and Thailand with respect to hydrological changes in the Mekong River, where geographic environmental factors play an important role, while sound water management policies are conducive to improving farmers' livelihood vulnerability, but poorly designed policies can also exacerbate vulnerability [51]. Especially for rice farmers in Vietnam, rising temperatures, drought, water pollution, soil erosion and river-

bank erosion are several factors that affect farmers' livelihoods, with rising temperatures and frequent droughts being particularly problematic [52]. At the same time, rice farmers are generally reluctant to change their cropping cycles in the absence of incentives and will be constrained by factors such as unstable rice markets and labor shortages in response to flood disasters [53], while farmers with a mixed rice-shrimp model have the greatest livelihood capacity to cope with climate change [54].

Hydropower development in the LMRB has great potential to enhance the sustainable livelihoods of farmers. In terms of direct effects, hydropower development will not only improve the hydrological environment and fishery output, but also provide grid services for production and living in surrounding areas. In terms of indirect effects, the construction of water and grid projects and transportation infrastructure will significantly improve the electrification technology and equipment conditions for the development of modern agriculture in the surrounding areas, and with the addition of appropriate agricultural policy subsidies and technical training, it will significantly improve the livelihood skills of the surrounding farmers and help them develop more income channels. At the same time, the dam's ability to regulate water resources during the dry and rainy seasons also helps mitigate farmers' losses from natural disasters.

However, hydropower development is a long-term, systemic project that will also have long-term impacts on the sustainable livelihoods of farm households. Although existing studies have identified many benefits and listed possible negative impacts of enhanced hydropower development in the LMRB, such as inundation of farmland, population displacement, river erosion, loss of biodiversity and ecological damage, there is insufficient understanding of the long-term impacts on farmers' responsive behavior and feedback effects, insufficient investigation of possible changes in farmers' perceptions and behaviors in hydropower development. This will make it difficult for the benefits of hydropower development to be recognized by the surrounding farmers and will hinder the development of hydropower development in the basin in the long term.

## 4. Exploring the Energy-Water-Food Nexus Analysis Framework and Its Application in the LMRB

The energy-water-food nexus analysis framework has become a research paradigm integrating multi-scale perspectives, such as exploring policy conflicts and coordination among energy security, water security and food security at the macro level, and exploring collaborative strategies for improving resource efficiency and livelihoods of residents at the industrial and residential levels, and focusing on portraying the dynamic interaction between top-down and bottom-up processes, which is an important reference for the exploration of medium- and long-term optimal sustainable development paths involving multi-levels, multi-sectors, multi-factors and multi-objectives.

### 4.1. Characteristic and Advantages of the Energy-Water-Food Nexus Analysis Framework

The nexus of energy, water, and food is a systemic expression of natural resource supply pressure and an essential element for improving human well-being and promoting poverty reduction and sustainable development, which is centered on trying to find the best trade-off between natural resource subsystems and socio-economic subsystems, and identifying the EWF coupling process is an important element for resolving multi-sector and multi-interest body nexus problems, mitigating conflict of interest and achieving a win-win situation [55].

Applying the energy-water-food nexus perspective and methodology, a wide range of studies have been conducted in various regions of the world, focusing on the interactions between energy, water and food-related sectors in a system, providing a systematic solution to the problem of conflicting interests and trade-offs among multiple factors and sectors as a "realistic option for achieving economic efficiency, resource efficiency and improved livelihoods" [56,57]. The main objective is to manage limited resources in an integrated manner to achieve a "green" economy that provides sufficient water, energy and food security for a

growing population [58]. At the same time, security issues and their challenges in the food, water and energy sectors are often intertwined in a variety of complex ways that cannot be effectively managed without cross-sectoral integration across basins. For example, in South Asia, the most striking feature of the energy-water-food nexus is the high dependence of downstream communities on upstream ecosystem services (dry season water for irrigation and hydropower, drinking water, soil fertility and nutrients) [59]. Livelihoods are a key component of achieving sustainable development, and the water-energy-food nexus framework should link the theory and practice of sustainable livelihoods to explore the balance between natural resource supply and human demand for the environment [60]. In response to climate change, in order to improve climate change adaptation and resource use efficiency, policy coherence should be enhanced in terms of linkage understanding to ensure integrated food, water and energy system security [61], minimize conflicts and maximize synergies between food-water-energy security [59]. Among them, the development of modern energy systems is a breakthrough to enhance the integrated sustainable development capacity of some regions [62].

*4.2. Analysis of EWF Nexus in Hydropower Development in the LMRB and Extension of the Framework for Farmers' Livelihoods*

Some progress has also been made in the study of the energy-water-food nexus for hydropower development in the LMRB. For example, one study argued that the construction of hydropower dams by Chinese SOEs in the LMRB lacked spatial assessment of the impact of the water-energy-food nexus and instead worsened the original silo management model, which led to the inability of local governments to establish effective communication channels with local communities [63]. Applying a water-food-energy nexus framework that portrays dynamic equilibria to the LMRB can explore the cross-sectoral evolutionary features of water, food and energy security and also discover the best implementation path for related policy support [64]. Meanwhile, a study on the collaborative effects of water-food-energy nexus in the LMRB based on a hydrological-economic optimization model found that dam operation could increase the output of irrigated crops by 49% with less impact on hydropower development and also reduce crop losses by 30% during the dry season, while enhanced environmental management could lead to a significant increase in fisheries output by up to 75%, but would also reduce irrigated crop yields by 48% and electricity output by 17% [65]. However, the existing research framework lacks a systematic consideration of farmers' livelihood issues. In the case of hydropower development in the LMRB, the inclusion of farmers' livelihoods in the energy-water-food nexus analysis framework not only helps to explore the benefits of hydropower development projects for farmers themselves and enhance their livelihoods, but also helps to assess the long-term benefits of hydropower development fully and accurately.

4.2.1. The Interaction Process between Hydropower Development and Farmers' Livelihoods in the LMRB

There is a systematic mechanism linking hydropower development and farmers' livelihoods in the LMRB. Farmers' livelihoods are centered on food production and need to coordinate the resource allocation and technical efficiency of water resources use, land use and energy use. Hydropower development is centered on sustainable water resource use and needs to weigh and coordinate conflicts between water resource use, land use, power deployment and peripheral irrigation. In terms of farmers, food production depends on the deployment of soil and water resources in the LMRB, while agricultural products are an important source of fuelwood for farmers in some countries and an important raw material for developing biomass, which in turn partly requires water resources in the basin, while changes in crop types will also have a significant impact on this process. Hydropower development needs to guarantee a stable supply of water resources, while climate change and ecological issues, as well as the recognition of hydropower development projects by neighboring farmers, will generate many uncertainties and risks (Figure 2). At the same time, the development of hydropower with the development of river flow monitoring tech-

nology and telemetry can effectively prevent flooding, thus reducing damage to agriculture and infrastructure, allowing farmers to produce rice in flood-prone areas and increasing the area under cultivation for local farmers [66–69]. According to some studies, during droughts, if reservoirs prioritize irrigation over hydropower, releasing more water during months of high irrigation demand (April and December) can reduce crop losses by up to 30%, significantly improving crop yields [66]. The construction of hydropower will also lead to improvements in the rough production structure in the LMRB, as the local government encourages rice production, dryland crops, cattle raising and commercial aquaculture to gradually replace traditional rough production methods, reducing wild fish catches and aquatic animals [68]. These resources, such as shrimp pond sludge and bagasse, can provide a richer source of fuel for local farmers [70]. In terms of food, nearly 66% of the population in the Mekong River Basin works in fisheries-related production, and 60% of the local population's protein intake comes from fish [71]. The reduction of local migratory fish populations due to the creation of dams has affected the primary fishing industry [13,72], and although aquaculture is gradually being promoted by the government, it is an important source of protein for the local poor. Hydroelectricity provides energy for mechanized equipment such as combines and rice milling machinery, which have greatly increased local food productivity in the past [73]. The coordination between upstream and downstream will likewise optimize the downstream irrigation as well as cultivation sectors. The coordination between upstream and downstream is better to help agricultural production against natural disasters [74,75]. Due to changes in land cultivation types, more arid crops are grown, which greatly reduces agricultural water use [71]. Different modes of hydropower operation can also have different impacts on EWFs. Dam operations can increase irrigation water use without significantly impairing hydropower production, increase irrigated crop income by 49%, and reduce crop losses during droughts by 30%. Second, eco-friendly management can increase fisheries production by up to 75%, but reduce irrigated crop production (−48%) and electricity generation (−17%) [32]. We should pay attention to the different impacts of hydropower construction on the production and livelihood of farmers in different regions of the Mekong River, and the different effects of the water-energy-food nexus, and the different effects of the EWF nexus relationship.

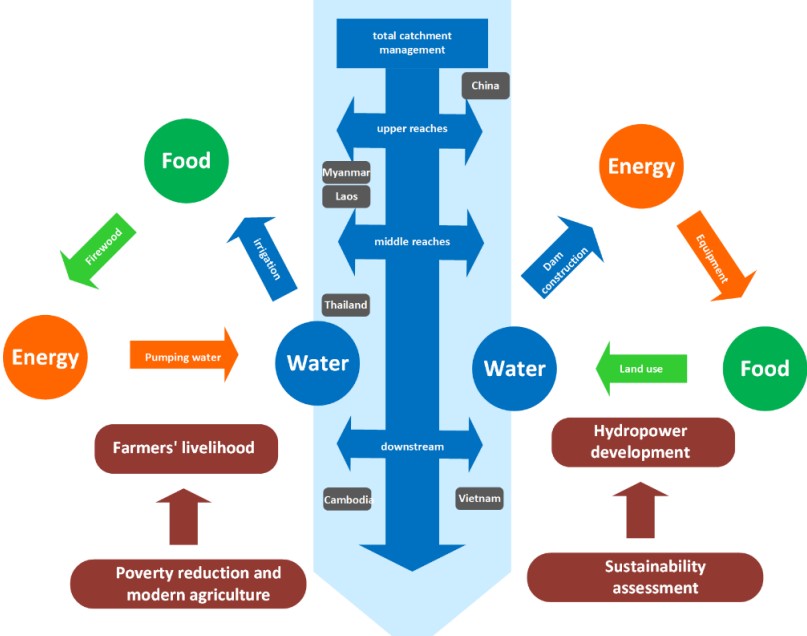

**Figure 2.** The interaction process of hydropower development and farmers' livelihood from the perspective of energy-water-food nexus.

4.2.2. An Integrated Indicator System of EWF Framework for Hydropower Development and Sustainable Livelihoods of Farming Households in the LMRB

Integrating the EWF framework for hydropower development with the indicator system for sustainable livelihoods of farmers can establish a comprehensive indicator system, both at the micro level and macro level. At the micro level, it can systematically understand the impact effects of the hydropower development process on food production, energy use and water use—especially the feedback effects brought about by the impact on the production and livelihood of farmers. Meanwhile, at the macro level, it can grasp the policy coordination process for food security, energy security and water security. Under the requirements of sustainable development goals, the comprehensive index system should focus on the current extremely urgent poverty reduction issues, such as improving the agricultural and rural market environment, increasing support for border trade and labor export, improving the production and livelihood security of farm households, strengthening government governance, easing the pressure on resources and the environment. However, it should also explore systemic optimization paths, such as strengthening infrastructure construction of transportation, electricity, trade, public health, improving the level of electricity coverage for farmers, improving the technical conditions of modern agricultural production and electrification of rural life, establishing a modern agricultural system, expanding diversified income channels for farmers and enhancing the ability of farmers to cope with climate change and natural disasters, so as to solve the long-term sustainability of farmers' livelihoods in the LMRB.

Figure 3 portrays the elements of the interaction between the EWF framework for hydropower development and the sustainable livelihoods of farmers, presenting the basic elements of the integrated indicator system.

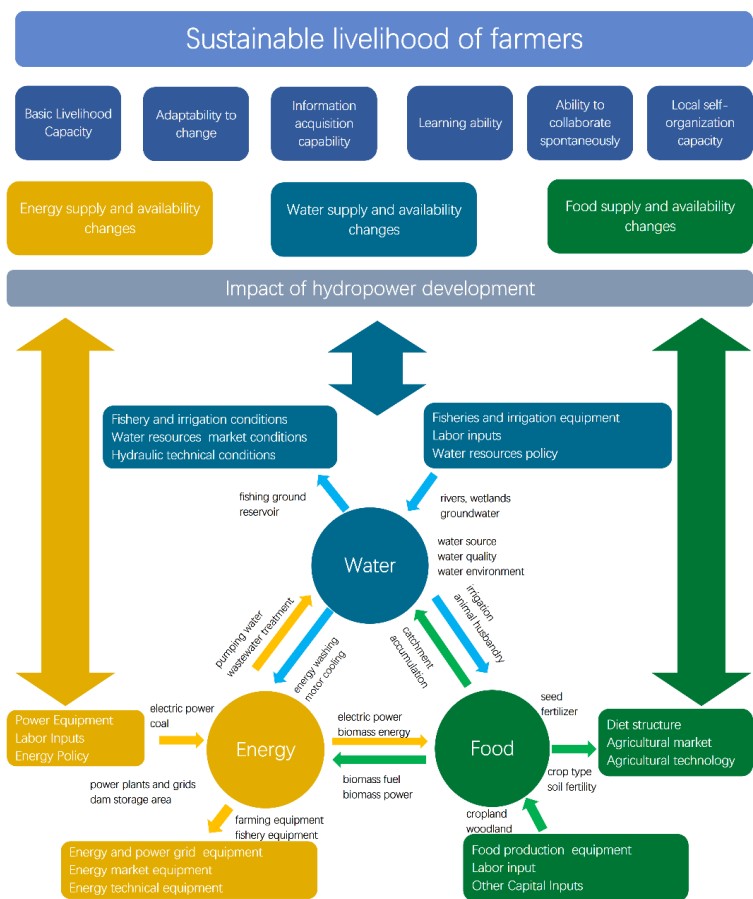

**Figure 3.** Interaction factors between hydropower development and farmers' livelihood from the perspective of energy-water-food nexus.

In terms of sustainable livelihoods for farm households, a survey of farm households in the Lancang-Mekong River Basin needs to be integrated. The main survey components are basic livelihood capacity, adaptability to change, information acquisition capability, learning ability, spontaneous collaboration and local self-organization capacity. The survey focuses on the impact of technological advances in hydropower, optimal allocation of energy and water resources on the livelihoods of farm households and their choices of response behaviors (Figure 3). The survey program is divided into five parts: first, the situation of basic livelihood conditions such as energy availability, food and drinking water security, transportation accessibility, disaster response, community environment and government subsidies related to farmers' livelihoods; second, the impact of changes in energy technologies and structures; third, the impact of changes in water resources technologies and allocation; fourth, the impact of changes in food storage and transportation conditions, with a focus on agricultural and fishery production; and fifth, the impact of the above changes on the basic livelihood conditions of farmers, the main methods of coping by farmers and the support and improvement programs that the government can provide (Figure 4).

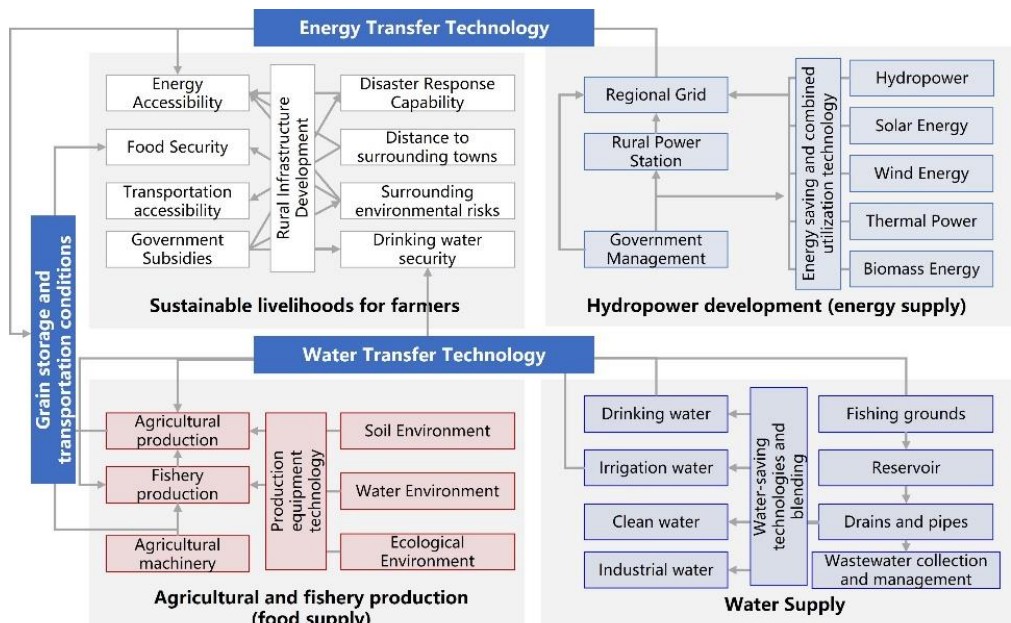

**Figure 4.** Sustainable livelihood survey program for farmers in the Lancang-Mekong River Basin, with concern on technological progress in hydropower and its impact evaluation.

### 4.2.3. The Basic Technical Framework of Coupling Model of Hydropower EWF and Farmers' Livelihood in the Lancang-Mekong River Basin

Based on the construction of a comprehensive indicator system, a coupling analysis model of the EWF nexus process of hydropower development and sustainable livelihoods of farmers (Hydropower EWF-Farmers' Livelihoods Coupling Model) will be applied to the coupling study of hydropower development and sustainable livelihoods of farmers in the LMRB, and Figure 5 provides a basic technical framework. The coupled hydropower EWF-farmer livelihood model consists of several sub-models and analysis modules, mainly including hydrology-water resources management model, economic-technological assessment model, farmers' response behavior, livelihood change module and natural environment model, which are integrated using a system dynamics framework.

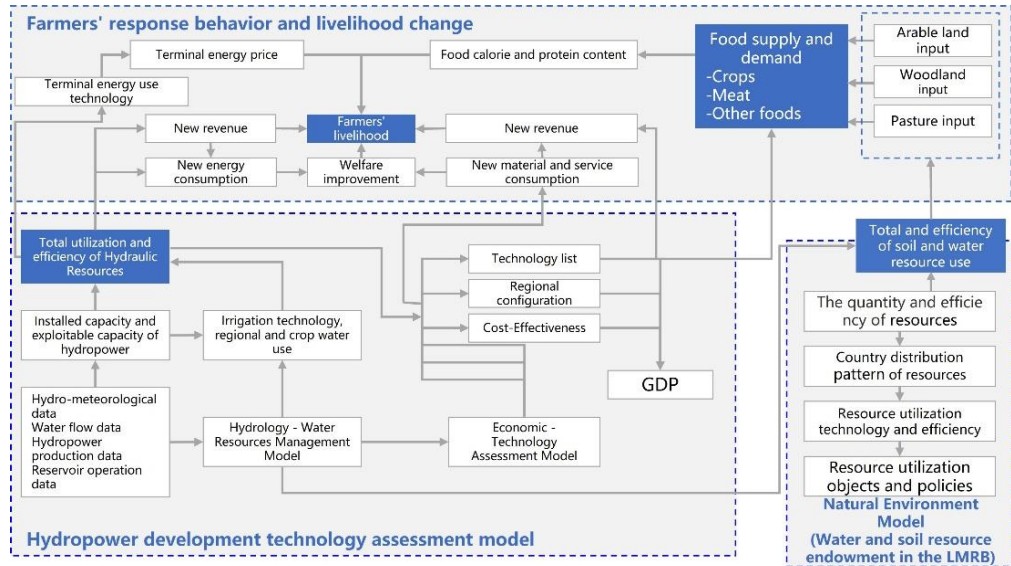

**Figure 5.** The basic technical framework of coupling model of hydropower EWF and farmers' livelihood in the Lancang-Mekong River Basin.

The economic-technical assessment model is an optimization model that aims to minimize the cost of generating and supplying electricity by building, improving or scrapping existing hydropower projects. It is based on the need to address climate change and sustainable goals and set a roadmap for hydropower development, power storage and grid transmission at different scales in different regions in the short and long term. The economic-technological assessment model optimizes long-term investments and short-term operations across multiple technology applications and portfolios, subject to reliability, operational and resource availability constraints, as well as current and potential future climate policies and environmental regulations. The key cost control objectives include investment costs for existing and new projects, fixed operating and maintenance costs for existing technologies, investment costs and expected returns of emerging technologies in the foreseeable future planning period, investment costs for new and existing transmission lines and distribution infrastructure and annual operating costs for new and existing transmission lines and distribution infrastructure.

The hydrology-water resources management model and the natural environment model describe the water and soil resource endowments in the LMRB and the processes of change they undergo because of climate and human activities, and influence hydropower development and farm household behavior models. The natural environment model will combine the application of spatial statistical analysis and GIS pattern analysis methods to quantify and analyze the total water and soil resources and exploitable quantities in the LMRB, portray the country distribution pattern, average technology and its efficiency level, stakeholders and policy interventions.

Farmers' response behavior and livelihood change analysis module. Farmers require a stable and growing source of income to maintain basic food, energy, water needs and access to modern services to support them. Based on the farmer questionnaire survey, the grid-based data analysis module is established by overlaying land use data, water supply data, climate change data and renewable energy use and distribution data through GIS technology. For example, calculate different grid energy supplies (electricity penetration = population/total population, electricity reliability = hours of reliable electricity service/total hours), water availability and consumption (water stress index = water consumption/water availability), food production and food consumption (food availability index = average daily nutrients in food harvest/recommended daily nutrient intake) to determine regional and EWF thresholds to provide basic EWF services to households.

## 5. Discussion and Suggestions

This study explores the mechanism of human-land system interactions and its energy-water-food coupling process in the interrelated issues of hydropower development and farmers' livelihoods in the LMRB and establishes an analytical framework based on the literature with the following theoretical values and practical implications.

First, theoretically, we can systematically characterize the coupled energy-water-food process in a typical watershed, deconstruct the mechanism of the linkage between hydropower development system and farmers' livelihood system, discover the key links and main processes and identify the linkage effects between optimal water resources allocation and sustainable livelihoods of farmers in response to global climate change.

Second, from a practical application perspective, we focus on assessing the current status and development potential of hydropower utilization across the basin by identifying the key factors of energy-water-food coupling in hydropower development in the LMRB and analyzing the path of hydropower technology advancement. The integrated energy-water-food security capacity and technical solutions to cope with climate change and improve farmers' livelihoods are combined with multi-scale data integration, system modeling and multi-country farmer studies to provide a scientific basis for planning and decision-making related to hydropower development in the LMRB for governments, enterprises and farmers.

In addition, for the countries in the LMRB, identifying and accurately quantifying the collaborative effects of hydropower development and sustainable livelihoods of farmers, and designing various policy options and technical choices based on the differences in regional environmental endowments and the characteristics of farmers' livelihoods, can provide scientific references for policy formulation in water resource development and management, poverty reduction and sustainable livelihoods of farmers in the basin countries.

## 6. Conclusions

This study presented the geographic and hydrological characteristics of the LMRB and explored how hydropower development in the Lancang basin has the advantage of high annual runoff, which can mitigate the risk of local natural disasters and enhance the economy by improving the natural conditions for local production. However, some negative impacts occurred on local farmers' livelihood, such as reduced fish catch and land salinization downstream.

The perspective of the energy-water-food nexus led us to better couple the livelihoods of local farmers and to assess hydropower development more comprehensively in the LMRB with a systematic framework. Based on the literature survey, we developed a collaborative analytical framework for hydropower development and sustainable farmer livelihoods with concern for the needs of energy, water and food security. Moreover, this framework includes a coupling model structure, which has three modules, including economic-technological assessment, hydrology and water resource management, farmer response behavior and livelihood change and the natural environment. The modules are linked from the uses of energy, soil and water resources, the supply and demand of food, the natural environment and resource endowment.

We believe that this collaborative analysis framework can provide a scientific basis for the planning and decision-making of governments, enterprises and farmers related to hydropower development in the LMRB. In the next step, we will apply this framework with the couple modeling to evaluate the impact of several hydropower projects with different dam operation modes and provide scientific reference to the optimal policy mix for pursuing synergistic optimization effects in water resource development and management, poverty reduction and sustainable livelihoods of farmers in the LMRB countries. Other regions and countries can also refer to this study, especially for those with a similar dilemma between hydropower development and farmers' livelihood improvement.

**Author Contributions:** This paper is a joint effort by several authors. Conceptualization, S.Z., Y.Z. and J.Z.; methodology, S.Z.; validation, L.S. and J.Z.; data collection and processing, S.Z. and Y.Z.; writing—original draft preparation, S.Z.; writing—review and editing, S.Z. and Y.Z. All authors have read and agreed to the published version of the manuscript.

**Funding:** This research was funded by the Second Tibetan Plateau Scientific Expedition and Research Program (STEP), Grant No. 2019QZKK1003, the International Partnership Program of Chinese Academy of Sciences, Grant No. 131A11KYSB20170117, the Strategic Priority Research Program of the Chinese Academy of Sciences, Grant No. XDA19040102 and the National Natural Science Foundation of China, Grant No.42071281.

**Institutional Review Board Statement:** Not applicable.

**Informed Consent Statement:** Not applicable.

**Data Availability Statement:** Not applicable.

**Acknowledgments:** We also thank Ayman Elshkaki from the Institute of Geographic Sciences and Natural Resources Research, CAS, for providing us with the many constructive suggestions to improve the manuscript.

**Conflicts of Interest:** The authors declare no conflict of interest.

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
