# Peer review of "A Collaborative Framework for Hydropower Development and Sustainable Livelihood of Farmers in the Lancang-Mekong River Basin: A Review with the Perspective of Energy-Water-Food Nexus"

_water, doi:10.3390/w14030499_

Round 1

Reviewer 1 Report

The authors did a good work from an experimental point of view, and I recommend the article for publication after some minor revisions.

More specific:

L1: Is the work an article or a review?

L27: Use only keywords from the title. You need to change them.

L112: Figure 1 looks good, but some information is missing, e.g., names of different countries and major rivers (in English).

L177: Number reference missing.

Reviewer 2 Report

The presented article has been prepared at a high substantive and scientific level, based on a well-selected set of source materials.

The assumed research goal was basically achieved.

The results are well presented in 4 figures. However, fig. 3 is hardly legible.

The conclusions are too general.

They should answer three questions posed by the authors at the beginning of the article (lines 136-133). Then they will correspond well with the title and content of the entire article. 
